# Electropolymerised pH Insensitive Salicylic Acid Reference Systems: Utilization in a Novel pH Sensor for Food and Environmental Monitoring

**DOI:** 10.3390/s22020555

**Published:** 2022-01-11

**Authors:** Monica Miranda Mugica, Kay Louise McGuinness, Nathan Scott Lawrence

**Affiliations:** ANB Sensors Ltd., 4 Penn Farm, Haslingfield, Cambridge CB23 1JZ, UK; mmugica@anbsensors.com (M.M.M.); kmcguinness@anbsensors.com (K.L.M.)

**Keywords:** square wave voltammetry, salicylic acid, reference electrode

## Abstract

This work summarizes the electrochemical response of a salicylic acid-based carbon electrode for use as a novel solid-state reference electrode in a redox-based pH sensor. This novel reference produces a pH insensitive response over a range of pH 3–10 in solutions with low buffer concentrations, different compositions, conductivities, and ionic strengths is produced. The pH of the local environment is shown to be determined by the chemistry and the electrochemical response of the redox active species on the surface of the electrode; the local pH can be controlled by the electropolymerized salicylic acid moieties due to the acid concentration on the surface, avoiding any perturbation in environmental pH and leading to a stable novel reference system. Sensitivities of −7.1 mV/pH unit, −2.4 mV/pH unit, −0.2 mV/pH unit, and 2.5 mV/pH units were obtained for different food medias, hydroponic solution, seawater, and cell-culture media, respectively, confirming its ability to control the local pH of the electrode. This reference system is paired with a new pH sensing element based on electropolymerized flavanone to provide a calibration free, pH sensitive sensor to effectively and accurately measure the pH of various media with high viscosity, low conductivity, low/high buffer concentration or cell-culture environment, presenting a maximum error of +/−0.03 pH units.

## 1. Introduction

The monitoring of pH is extremely important in many industries, such as pharmaceutical, food, agricultural and environmental. This is because a number of chemical processes are dependent on the hydrogen ion concentrations [1,2,3,4,5]. It is the most common analytical measurement in industrial processing, and it plays a very important role in the food processing industry in order to ensure reliable product quality and consistent, well-defined properties which meet regulatory requirements to avoid causing health problems for consumers [6]. Indeed, regulatory compliance is not the only reason why pH is important. The pH of the media impacts food characteristics such as texture, taste, aroma, and others. Effective monitoring of pH in the food industry begins with testing raw materials and continues throughout production to the finished product. Acidic food has a natural pH of 4.6 or below. When acid is added to lower the finished pH to below 4.6, these foods are known as ‘acidified foods’, which include canned goods, salsas, sauces, and fruits, among others. In the production of these, a pH higher than 4.6 will end in an environment that encourages the growth of unfavorable bacteria that can cause illness [7].

Determining best practices for measuring pH can be a challenge, and despite its importance, very few new technologies have entered the market, which has a heavy reliance on the electrochemical glass pH probe. A review of the literature shows the glass pH electrode is still used in most applications because the glass bulb is reliable, easy to calibrate, sensitive to pH and provides quick real time measurements. However, even though it is easily calibrated, the necessity for calibration due to a drift in the reference electrode means it has high operational costs due to the number of man hours involved in the process [8]. The drift is caused by changes in the environment of the reference electrode chamber, impacting the potential of the Ag/AgCl redox couple and therefore leading to an inaccurate pH measurement [9,10]. Further issues arise when trying to accurately measure pH in foods with high viscosity. Because of the limited interaction between the aqueous electrolyte and the viscous sample, there are a lot of inconsistencies in the reading and a longer response time.

Glass electrode manufacturers and researchers alike have endeavored to overcome reference electrode drift; focusing on improving the stability of the reference electrolyte to minimize ion mobility and exchange, improving the performance of the porous frit, or introducing junction systems. Solid-state reference electrodes (SSRE) have emerged as an alternative for pH insensitive electrodes due to the simplicity and ease of fabrication reducing the problems associated with conventional reference electrodes. However, they suffer drift when placed in complex media [11,12,13].

Pioneering work by Hickman et al. [14] demonstrated that conventional reference electrode limitations may be overcome by measuring the potential difference between the response of a pH sensitive and a pH insensitive (no proton transfer) redox compound. However, although such systems produce a pH insensitive signal, it is often found that the redox potential of the species is dependent on the concentration of ions in solution. A significant amount of focus has been on the use of ferrocene-based species, which experience a one electron oxidation to the resulting ferricenium ion [15,16]. Nevertheless, ferrocene has shown to suffer from electron mobility issues through the ferrocene chain, and the ferrocenium cation suffers from chloride-induced disproportionation [17]. This is therefore of no use for tracking reference potential voltage within systems containing unknown and changing concentrations of ions. The challenge, though, is to find a stable reference compound capable of being insensitive to varying concentrations of species found within the solution being tested.

Electrochemical system conditions are strongly dependent on the ionic strength of the solution [18]. M. Quan et al. [19] demonstrated the difference in redox behavior of quinones in buffered and unbuffered media, which is explained by the altered conditions in non-buffered media due to the voltammetry of redox species involving proton transfer. The electrochemical process involves consumption or release of protons at the surface of the electrode, perturbating the local pH of the solution, thus causing erroneous behavior. A simulation model presented by Compton et al. [20] shows that in some conditions the pH at the electrode surface can be altered up to 5–6 pH units comparing with that of the bulk solution. Compton noted that redox active species control the environment local pH of an electrode, but the difficulty of measuring the pH in that low buffered systems persisted. With the aim of monitoring the local pH in low ionic strength media, Dai et al. [21] presented a solution by manipulating the structure of the compounds on the electrode surface and proposed modified anthraquinone structures which facilitate proton transfer between the water molecules and the redox active moiety [22].

Lawrence et al. [23] utilized this local environment control to produce a new reference system in which the electrochemical process provided control of the pH in the environment local to the surface of the electrode. It has been seen that surface functional groups fundamentally affect adsorption characteristics of the material and so electrochemical performance [24].

This work presents a novel salicylic acid-based solid-state electrode for use as a stable voltametric reference. Salicylic acid (SA) presents a dissociation constant (pK_a_) of 2.97 and can be electropolymerized to obtain an electrochemical signal dependent on the pH of the system [25]. However, through impregnation of salicylic acid into a carbon epoxy electrode followed by subsequent electropolymerization, it is found that the acid moiety effectively sets the pH, at the pK_a_ value, in the vicinity local to the electrode. This paper demonstrates the versatility of this reference electrode through different medias, from high viscosity food-media to low buffer and low conductivity conditions, to high buffer and bio-based environment. Furthermore, Nafion^®^ coated, electropolymerized, pH sensitive 2′-Hydroxyflavanone-based polymer is presented as a pH sensor [26], whereby Nafion^®^ is used as a proton-exchange membrane to encourage the proton transfer between the media and the electropolymerized polymer.

This voltametric pH sensor is demonstrated in semi-solid food medias, such as soy sauce and raspberry syrup, in a hydroponic nutrient solution (Hoagland), sea water, and high buffered cell-culture media (DMEM), as well as standard buffer solutions. Hoagland solution provides every essential nutrient required by green plants and is appropriate for supporting growth of a large variety of plant species [27].

## 2. Experimental

### 2.1. Apparatus

Electrochemical measurements were conducted using an Ana Pot potentiostat (Zimmer & Peacock, Coventry, UK) with a standard three-electrode configuration. A flavanone and salicylic acid-based carbon composite electrode were used as the working electrodes, a carbon counter and an Ag/AgCl (BASi, West Lafayette, IN, USA) or a silver wire acted as the reference electrode.

Absolute pH measurements were performed using a standard glass electrode (Sensorex, Garden Grove, CA, USA). Prior to the measurement of the medias, the pH meter was calibrated using Reagecon buffers of pH 4.01 ± 0.01, pH 7.00 ± 0.01 and pH 10.01 ± 0.01 (Reagecon Diagnostics Ltd., Shannon, Ireland). Measurement of the pH was carried out on each freshly made solution prior to experiments. All the experiments were carried out at 20 ± 1 °C.

### 2.2. Reagents

All chemicals were purchased from Sigma–Aldrich, Dorset, UK, and used without further purification (unless specified). Standard IUPAC buffer solutions (pH 4, 7, 9) were prepared as follows: pH 4.07—potassium hydrogen phthalate (0.05 M); pH 6.86—potassium dihydrogen phosphate (0.025 M) and sodium phosphate dibasic (0.025 M); pH 9.23—sodium tetraborate (0.05 M), all in deionized water (Hexeal, Brundall, UK). All buffers contained 0.1 M KCl as the supporting electrolyte.

Britton–Robinson buffer was used for the electropolymerization of salicylic acid (SA) carbon-based electrode, which consists of a mixture of 0.04 M boric acid, 0.04 M phosphoric acid and 0.04 M acetic acid.

Raspberry syrup and soy sauce were purchased from local groceries and used as food-media testing. For the pH calibrations, different concentrations of 0.5 M NaOH stock solution were added into the semi-solid food medias, and the corresponding pH was measured using the standard glass electrode.

Hoagland’s No. 2 Basal Salt Mixture was dissolved in 1 L deionized water and used as a hydroponic nutrient solution. Hoagland was diluted to 1:20, leading to a very low conductive solution. NaOH 0.5 M and HCl 0.5 M stock solution were used for adjusting the required pH.

Sea water, H2Ocean Natural Reef Salt, was purchased from Maidenhead Aquatics (St. Albans, UK) in which 1 kg of this salt was dissolved in 25 L of water. For the seawater calibrations, different concentrations of CO_2_ were bubbled into a stirred sea water solution, and the corresponding pH measured using the standard glass electrode.

Dulbecco’s Modified Eagle’s Medium (DMEM) was used as a high buffered bio-media (purchased from Sigma Aldrich, Dorset, UK). Each powder bottle (around 8.5 g) was dissolved in 1 L deionized water, and 3.7 g/L sodium bicarbonate and 0.0159 g/L phenol red was added to the solution in order to match the commercial composition of the bio-media.

### 2.3. Preparation of Carbon Composite Electrodes

The flavanone-based carbon composite electrodes comprised of multi-walled carbon nanotubes, 2′-Hydroxyflavanone, Nafion^®^ perfluorinated resin solution, with RX771C/HY1300 epoxy resin (purchased from Robnor ResinLab Ltd., Swindon, UK) used as the binder [26].

The salicylic acid-based carbon composite electrodes consist of COOH-functionalized graphitized multi-walled carbon nanotubes (30–50 nm outside diameter, purchased from Cheap Tubes, Cambridgeport, VT, USA), glassy carbon (purchased from Sigma Aldrich, Dorset, UK), salicylic acid as redox active species, and RX771C/HY1300 epoxy resin (purchased from Robnor ResinLab Ltd., Swindon, UK) as the binder.

The preparation of both pH insensitive and pH sensitive electrodes is very similar. The carbon composite was first ground in a mortar until the powder was fine enough to mix homogeneously. The pH sensitive/insensitive species were then added and ground with the carbon in a known weight-to-weight ratio. In the case of the Nafion^®^-coated electrodes, the flavanone compound was previously dissolved in Nafion^®^. Both mixtures were dried at room temperature and were added to the carbon mix. Once the mixtures were homogeneous, they were carefully mixed with the epoxy resin, to form a carbon epoxy paste.

Each resulting mixtures were then packed into a recess (5 mm length, 1 mm diameter) of a PEEK™ manufactured body. Electrical connection was made using brass rod (4 cm length, 1 mm diameter). The electrodes were cured at 125 °C for 1 h to produce the solid carbon composite electrode.

## 3. Results and Discussion

Electropolymerization of salicylic acid (SA) carbon-based electrodes were first achieved using square wave voltammetry (SWV) in Britton–Robinson buffer, which is reflected in Figure 1a. This presents the first and thirtieth SWV of the reductive polymerization of a salicylic acid electrode. Scan 1 exhibited a reductive wave at +0.755 V, which increased in magnitude with increasing scan numbers. Furthermore, a shift in the potential to +0.705 V was observed on scanning. This growth in peak current and shift in potential is consistent with the formation of a SA-based polymer on the electrode surface.

This result is consistent with the literature data which shows that oxidation of the SA monomer in acidic conditions produces a polymeric species on the electrode surface. Although the SA oxidation products have not been clearly identified, it is believed the oxidation is similar to that of phenol, as demonstrated by Park et al. [28] (Figure 1). In this process it is shown that long chains of electropolymerized SA are difficult to obtain, rather the electrode is coated with short chained polymeric species. Indeed Figure 1 shows the formation of three different radicals, forming some dimers after propagation, with a further oxidation of SA products. However, some trimers may also be formed during electropolymerization as well. Park et al. suggested that the electrochemical oxidation of SA can be impacted by intermolecular H-bonds formed by the carboxyl group, and intramolecular H-bonds with the closest phenolic functional group, C=O···HO. The electrochemical reaction of the electrode when submerged in a solution is based on the hydrogen bonding formed by the hydrogen of the acid group, forming carboxylate anions on the surface of the electrode. The carboxylate anion is stabilized by intramolecular hydrogen bonding with the adjacent phenolic group [29].

Once polymerized, the SA electrode was removed from the pH 2 polymerization solution and placed in IUPAC buffers, and the voltammetric response was analyzed. These data are shown in Figure 1b. Unlike the previous literature data [30], the peak potential of the redox active wave was found to be insensitive to the pH of the solution. This change in behavior was acquired by increasing the concentration of SA moieties in the surface of the electrode, and thus, increasing the ability of the chemistry to control the local pH. Knowing that the pK_a_ of salicylic acid is 2.97, it is assumed that the peak potential observed for the electrode in buffered media corresponds to pH 4, consistent with the pH at the surface of the electrode. This is accordant with a high concentration of the SA being present on the electrode surface and setting the environment locale to the electrode by SA acidic moieties. This phenomenon allows the possibility of such an electrode to be used as a voltammetric reference electrode within certain low buffered media similar to that found in a number of food stuffs.

In order to study the versatility of salicylic acid electrodes, SWV were performed in different solutions: food-medias (raspberry syrup and soy sauce), Hoagland solution, synthetic seawater, and Dulbecco’s Modified Eagle’s Medium (DMEM). On one hand, raspberry syrup, soy sauce and Hoagland solution were used to demonstrate the ability of salicylic acid electrode as a reference system in the food industry. Raspberry syrup and soy sauce were interesting to the authors especially due to the rheological properties they possess, and the current issues the glass pH meter presents in these high viscosity medias. Hoagland is a hydroponic solution which provides every nutrient necessary for plant growth, and supports the growth of many plants. This solution was diluted in 1:20, in order to get a very low conductive solution (around 2 mS/cm). On the other hand, salicylic acid-based electrode was also tested in synthetic seawater and DMEM bio-media, in order to show the adaptability of the pH insensitive sensor in different conditions, such as high salinity or different buffer concentrations, or even in a cell-culture media.

SA electrode was first tested in both raspberry syrup and soy sauce, which had a measured pH 3.18 and 3.60, respectively, using a standard glass electrode. It can be foreseen that in a real-world application the SA electrode may not be polymerized prior to being tested so initial tests focused on understanding if the electropolymerization of SA would occur in this media. The electrode was first abraded to ensure a fresh, unpolymerized SA surface was present. The electrode was then immersed in soy sauce and 30 repetitive square wave voltammograms were recorded. Figure 2a shows the first and thirtieth voltammetric responses of the electrode. It can be clearly seen that the result is analogous to that shown in Figure 1a, with well-defined voltammetric responses observed and a stable peak potential from +0.701 V to +0.695 V over the thirty scans. This indicates the SA polymer can be easily formed in this media.

To ensure the SA electrode exhibited pH insensitivity, and hence could be used as a reference electrode in this high viscosity media, 0.5 M NaOH aliquots were added to the soy sauce. The resulting voltammograms are shown in Figure 2b, confirming the stability of the peak potential over different pH values. Figure 2b insert shows the resulting plot of the peak potential as a function of pH for soy sauce and pH 4, pH 7 and pH 9 standard buffers. This shows a sensitivity of −8.72 mV/pH unit for standard buffers, presented by the black trendline, and a sensitivity of −7.47 mV/pH unit for soy sauce. This is in contrast with the previous literature data [30] and confirms that by increasing the concentration of the SA at the surface, the sensitivity of the SA electrodes is close to zero due to the carboxylic acid functionalities setting the local pH of the electrode to its pK_a_.

The SA electrode was then tested in 1:20 diluted Hoagland solution. It was first abraded, removing the polymer formed on the surface of the electrode. SA was then polymerized directly in Hoagland solution (pH 7.17). The pH was then adjusted by adding 0.5 M NaOH and 0.5 M HCl stock solution additions providing a pH range from 3.07 to 9.30. Well defined square wave voltammograms were obtained, as shown in Figure 3a in all solutions. The resulting plot of peak potential as a function of pH for the SA-based electrodes in diluted Hoagland solution overlayed the ones for raspberry syrup, soy sauce and IUPAC standard buffers, as detailed in Figure 3b. The sensitivity obtained for Hoagland solution was −2.44 mV/pH unit, confirming the ability of the electrode to control the pH local to the electrode surface. These data thereby confirm the purpose of using SA-based carbon electrodes as a reference electrode in high viscosity food media as well as in hydroponic solutions with very low conductivity. The average peak potential of the SA electrodes in a range of pH 2.83–9.30 for both medias was 701.9 ± 7.2 mV.

In order to prove the adaptability of the SA electrode as a pH insensitive sensor in different conditions, the electrode was tested in synthetic sea water, and in DMEM high buffered bio-based solution. In this case, the electrode was abraded again to ensure a fresh, unpolymerized SA surface, and electropolymerized in Britton–Robinson for 30 scans (not shown) prior to each test. Once polymerized, SA electrode was tested in sea water, with additions of different CO_2_ concentrations for adjusting the required pH. The pH measured by the glass pH probe ranged 8.20 to 6.51. Additions of NaOH 0.5 M and HCl 0.5 M stock solutions were added to DMEM to adjust the pH across the range from pH 6.66 to pH 9.42. The results obtained for the calibration plot of SA electrodes in seawater/CO_2_ system and DMEM solutions overlayed the previous food media and hydroponic solutions, with an average peak potential of +709.2 ± 0.4 mV and +699.3 ± 6.4 mV, subsequently, as shown in Figure 4. The sensitivities exhibited in these two medias were −0.25 mV/pH unit for seawater and +2.50 mV/pH unit for DMEM. Sensitivity being very close to zero confirms the purpose of using salicylic acid carbon-based electrode as a reference electrode in low buffered systems, indicating the ability of the chemistry to control the environmental local pH of the electrode surface, due to the acid concentration of the surface.

It has been seen that salicylic acid-based carbon electrode can be used as a reference electrode, since the electrode is able to control the local pH without perturbating, leading to a stable peak potential over all the solutions used, without regarding the composition, ionic strength, or conductivity.

With the aim of using SA electrode as a reference system, the electrochemical response of the flavanone-based pH sensing electrodes was then tested using square wave voltammetry in the same medias, and the calibration plot obtained is also presented in Figure 4, showing a pH sensitive behavior. The flavanone electrode was first electropolymerized in the food media as outlined previously [19]. In this case a reductive wave was observed at +0.465 V for scan 1 which was shifted to +0.440 V on consecutive scans and stabilized after 5 scans (not shown). As with SA electrodes, pH was adjusted in each solution to study the behavior of the electrode for different pH in each media. Figure 5 presents the redox waves for each solution. It can be clearly observed that the peak potential shifts to lower potentials as the pH of the solution increases. Black trendline in Figure 4 (circles) shows the buffer calibration of Flavanone electrodes, with a sensitivity of −55.7 mV/pH. A linear plot of peak potential with pH was observed in all the different conditions, overlaying the buffer calibration trendline with a sensitivity of −46.2 mV/pH, −53.6 mV/pH unit, −54.8 mV/pH unit and −55.8 mV/pH unit for food media, Hoagland hydroponic solution, seawater/CO_2_ system and DMEM bio-based media, respectively. The resulting polymer was found to provide a Nernstian response in high/low buffered, low conductivity, high viscosity, and cell-culture medias.

Figure 6 shows the results when this flavanone-based pH sensing electrode is combined with the SA-based carbon reference electrode. Subtracted values were calculated using the calibration trendline of SA electrodes over the flavanone’s calibration line. As expected, peak potentials obtained in all the different solutions follow the same trendline as the IUPAC buffers calibration, meaning that the SA reference electrode is controlling the pH of the local environment, regardless of the composition, ionic strength, conductivity, or rheological properties of the environment.

Table 1 summarizes the data obtained from each calibration plot, excluding IUPAC buffers. The pH was accurately calculated in all the solutions, demonstrating the ability of a flavanone/SA voltammetric-based sensor to monitor pH medias well known to be troublesome for the traditional glass electrode such as high viscosity solution, low buffered, low ionic strength, hydroponics, and cell-culture media.

Finally, to demonstrate the efficacy of SA pH insensitive sensor as a long-term pH sensor, experiments in soy sauce and diluted Hoagland solution were carried out. The two environments were chosen specifically as one exhibits high viscosity (soy sauce) and the other is a low conductivity and low buffer solution (diluted Hoagland hydroponic solution), both of which are very interesting to the author.

Figure 7 shows the SWV of SA electrode every 20 tests (which corresponds to 60 scans) in soy sauce. The electrodes were immersed and scanned every half an hour over 10 days, with a total of 260 measurements. They showed a slight decay in signal over the first three sets, and an increased in stability thereafter. After 10 days of testing, the electrode still showed a very defined SWV, verifying its long lifetime capabilities and long-lasting accuracy in high viscosity media.

Long term experiments in Hoagland 1:20 diluted were also carried out. SA electrode was first abraded to ensure an unpolymerized surface and polymerized in Britton–Robinson solution. It was then stored in Hoagland x20 solution over a week, in order to study the influence of the solution in the calibration plot. The storage condition did not affect the chemistry of the electrode, as Figure 8 presents. A sensitivity of 0.18 mV/pH unit as recorded after day 5 with the calibration plot overlaying that obtained on day 1. The SA electrode was immersed and scanned every hour over 37 days, with a total of 5000 scans showing the long-term stability of the electrode in these media.

This successfully long-term data confirmed the ability of salicylic acid moiety to behave as a reference electrode for low buffer systems, showing very stable potentials over a long period of time, making it suitable for deployment with no calibration needed.

## 4. Conclusions

A novel solid-state reference electrode utilizing salicylic acid carbon composite-based electrodes has been successfully developed, which overcome the issues of standard reference electrodes. The resulting polymer was found to provide an insensitive response in high viscosity food media, low conductivity and low buffer hydroponic solution, sea water, and high buffered cell-culture media, presenting sensitivities of −7.1 mV/pH unit, −2.4 mV/pH unit, −0.2 mV/pH unit, and 2.5 mV/pH units, respectively, and thus, indicating the ability to control the pH of the local environment around the electrode surface, due to the acid concentration on the surface. SA reference electrodes were successfully tested in solutions with different compositions, ionic strengths, and conductivities.

Flavanone-based pH electrodes were successfully tested in all the solutions used for SA measurements, providing a Nernstian response by combining with the SA reference electrode, demonstrating that the SA electrode can be efficiently used as a reference system. The calculated pH values were in excellent agreement for each solution, showing a maximum error of +/−0.03 pH unit, based on the equation obtained from the calibration plots and the measured pH using a calibrated glass electrode.

The efficacy of SA pH insensitive sensor as a long-term reference for a pH sensor was demonstrated in soy sauce and diluted Hoagland solution. After 10 days of testing in soy sauce, the electrode still showed a very defined SWV with no drift in potential, verifying its long lifetime capabilities and long-lasting accuracy in high viscosity media. It was also demonstrated that the sensor could be stored in Hoagland solution (1:20) for over a week without adverse effect. A sensitivity of 0.18 mV/pH unit was shown after a week, with the calibration plot overlaying that obtained on day 1 (−2.4 mV/pH unit). The SA electrode was immersed and scanned every hour over 37 days, with a total of 5000 scans confirming the long-term stability in these media.

## Data Availability

Not applicable.

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
