# Peer review of "Electropolymerised pH Insensitive Salicylic Acid Reference Systems: Utilization in a Novel pH Sensor for Food and Environmental Monitoring"

_sensors, 2022, doi:10.3390/s22020555_

Round 1

Reviewer 1 Report

This is an excellent manuscript which describes an elegant solution for monitoring pH.  The work comprises an highly innovative approach to pH sensing through the use of immobilised molecular reagents, and which demonstrate accuracy and very good long-term stability using the authors novel reference electrode.

This manuscript should be accepted in its current form - I only spotted four typographical errors that could be corrected at the proofs stage:

  1.  Page 2, third paragraph, first line:  reference 14 (Hickman et al.) should be reference 11.
  2.  Page 2, third paragraph, seventh line:  I prefer the use of the term "ferricenium", since ferrocene has iron formally in the +2 oxidation state and the cation has iron in the +3 oxidation state.
  3. PAge 3:  penultimate paragraph of section 2.2:  "kg" not "Kg".
  4. Page 4:  second paragraph of section 3:  "three" not "3".

Reviewer 2 Report

The authors reported an electropolymerised pH-insensitive salicylic acid reference system for food and environmental monitoring. However, the authors had previously published a work regarding a pH sensor based on electropolymerized substituted-phenol, such as salycilic acid, for pH detection in unbuffered systems (reference number 30 -DOI: 10.1039/c5ra22595g). It is not clear the difference between the new approach and the previous one. The authors wrote the following sentence: "Unlike previous literature data [30] the peak potential of the redox active wave was found to be insensitive to the pH of the solution." This behavior must better discussed. What are the differences between both approaches?

Please, find below other remarks:

1) In the following sentence: "However, even though it is easily calibrated, the necessity for calibration due to a drift in the reference electrode means it has high operational costs [8]." I do not see the calibration as a costly procedure, please clarify it. The calibration is an extra step, it could be time-demanding, but not costly.

2) The reference number 26 is not avaliable since it is a submitted paper. The authors used this reference to justify the observed behavior. For example: "Flavanone based pH electrodes [26] were successfully tested in all the solutions used for SA measurements, providing a Nernstian response by combining with the SA reference electrode, demonstrating that the SA electrode can be efficiently used as a reference system." I do not recommend its citation since we cannot check it yet.

3) Since the electropolymerization of salycilic acid is not fully understood, it is important to provide more data about it. It would be interesting to investigate the performance of the electropolyrimerized electrode through electrochemical impedance spectroscopy (EIS) after one scan and after 30 scans. EIS data should be compared to the non-modified electrode as well.

4) In the following sentence: "This is in contrast with the previous literature data and confirms that by increasing the concentration of the SA at the surface, the sensitivity of the SA electrodes is close to zero due to the carboxylic acid functionalities setting the local pH of the electrode to its pKa." Which is the previous literature data? 

Reviewer 3 Report

Manuscript submitted by Monica et al is well presented. However, I could not see much novelty in this paper, since most of the materials, reagents and electrodes are commercially procured.

Though, electropolymerization was done for salicyluc acid. I would like to see measurement of pH with and without SA polymerization to compare the performance.

Also, authors should add a comparative table with current state of the art methods in similar domain.

I strongly suggest to add more analytical data in abstract and conclusion along with the novely claim in the current work.

Round 2

Reviewer 2 Report

All the corrections/suggestions were provided. I do recommend it publication.

Reviewer 3 Report

Authors have incorporated suggeated changes and justified the questions raised by me. This manuscript now be accepted for publication.